# RNA-Seq Analysis of Prickled and Prickle-Free Epidermis Provides Insight into the Genetics of Prickle Development in Red Raspberry (*Rubus ideaus* L.)

**Archana Khadgi and Courtney A. Weber ***

School of Integrative Plant Science-Horticulture Section, Cornell AgriTech, Cornell University, 630 W. North St., Geneva, NY 14456, USA; ank53@cornell.edu
* Correspondence: caw34@cornell.edu

**Abstract:** Red raspberry (*Rubus idaeus* L.) is a globally commercialized specialty crop with growing demand worldwide. The presence of prickles on the stems, petioles and undersides of the leaves complicates both the field management and harvesting of raspberries. An RNA sequencing analysis was used to identify differentially expressed genes in the epidermal tissue of prickled "Caroline" and prickle-free "Joan J." and their segregating progeny. Expression patterns of differentially expressed genes (DEGs) in prickle-free plants revealed the downregulation of some vital development-related transcription factors (TFs), including a MIXTA-like R2R3-MYB family member; MADS-box; APETALA2/ETHYLENE RESPONSIVE FACTOR (AP2/ERF) and NAM, ATAF1/2 and CUC2 (NAC) in prickle-free epidermis tissue. The downregulation of these TFs was confirmed by qRT-PCR analysis, indicating a key regulatory role in prickle development. This study adds to the understanding of prickle development mechanisms in red raspberries needed for utilizing genetic engineering strategies for developing prickle-free raspberry cultivars and, possibly, other *Rubus* species, such as blackberry (*Rubus* sp.) and black raspberry (*R. occidentalis* L.).

**Keywords:** spine-free; thornless; RNA-Seq; glandular trichomes

## 1. Introduction

A plant's epidermis plays an important role in protection/defense against pathogens and predators. It acts as an active interface with the environment, controlling the vital exchange of gas, water and nutrients. Epidermal structures like the cuticle, hairs and trichomes, as well as waxy exudates, provide protection to the plants in many ways [1]. Plant trichomes are simple hair-like structures that are present in most terrestrial plants and are an extension of the epidermal surface [2]. Trichomes increase the effective thickness of the epidermis and help create a barrier between the epidermis and the environment, which moderates surface temperatures and helps reduce the transpiration rate [3]. Trichomes also provide some protection against biotic stresses like herbivores, insects and pathogens, as well as mechanical injuries [4–6]. Additional epidermal outgrowths such as thorns, prickles and spines provide additional mechanical protection from herbivory and mechanical damage. Thorns, prickles and spines are similar structures with different biological features that are, however, all commonly referred to as "thorns". Botanically, thorns are modified branches, and spines are modified leaves, both containing vascular tissues. Prickles, on the other hand, are an outgrowth of the epidermis formed by multiple cellular divisions and lack vasculature [7–11], thus having a biological similarity to trichomes.

*Rubus idaeus* L. (red raspberry), belonging to the Rosaceae family, is a globally commercialized specialty crop with a high fresh market value. Increased commercialization has been enabled by a rapid

increase in the cultivation of raspberries worldwide. Fresh market raspberries rely on hand-harvesting, with a corresponding high labor/expense input. While prickles help in defending against natural predators, these structures are an unappealing trait for domestication and commercial production [12] in *Rubus* and cause difficulties during cultivation, harvesting and field management. The development of prickle-free *Rubus* cultivars is desirable throughout the commercial industry, and breeders have made efforts in incorporating the prickle-free phenotype in their programs. Prickle-free cultivars such as "Joan J." and "Glen Ample" red raspberry [13] and "Natchez", "Chester" [14], "Apache" and "Triple Crown" blackberries (*Rubus* hybrid) highlighted the benefits to the industry. However, combining the prickle-free trait with other important traits through traditional breeding approaches can be time-consuming and expensive. Within *Rubus*, there are both prickled and prickle-free cultivars with similar genetic backgrounds, which aids in comparative studies to understand prickle development at the morphological and molecular level.

There have been several studies in understanding morphological structures, histochemical features, the origin and genetic patterns of prickles in Rosaceae species [15–18]. Morphological studies on understanding prickle development in multiple *Rubus* species suggests that prickles are modified glandular trichomes [16,18]. A similar association of glandular trichomes and prickle development is seen in *Solanum viarum* Dunal [19]. Unicellular simple trichomes in *Arabidopsis thaliana* (L.) Heynh. have proven to be great models for understanding the molecular processes behind cell fate and differentiation [20–22]. The members of the gene families MYB (myeloblastosis), transcription factor (MYB TF), basic helix-loop-helix (bHLH) and domain protein/WD-repeat (WD40) are known to play crucial roles in cell fate determination in unicellular trichomes [23–25]. However, there has been little progress in understanding molecular cues behind glandular trichome development [26,27]. A study on molecular mapping and candidate gene analysis for spines (non-glandular multicellular trichomes that were referred to as spines in the study) on cucumber (*Cucumis sativus L.)* fruits predicted a WD40 TF family gene to be responsible for trichome density [28]. Similarly, TRANSPARENT TESTA GLABRA1 in cucumbers (*Cs*TTG1), which is a homolog of WD40 repeats, was identified as the regulator of cucumber fruit wart/spine development, providing an insight into gene networks leading to the conversion of glandular trichomes into prickles [29].

In roses (*Rosa* species), a study to determine the genetic and molecular mechanisms of prickle development utilized an interspecific population that produced a low percentage of prickle-free plants [30]. QTL analysis agreed with previous studies in roses that suggested a recessive inheritance of the prickle-free trait [31,32]. Unfortunately, the phenotyping in this study was poorly defined, and examination of the morphology in the study suggests the complicating factor of glandular hairs rather than prickles, which is controlled by a different locus in *Rubus* [33]. The presence/absence of prickles in roses were mapped to a major recessive locus with three minor QTL related to prickle density also found, similar to what was found in red raspberries [34–36]. The interspecific nature of the population could account for some of the anomalous phenotypes and poor fit to expected inheritance patterns. Unfortunately, it seems this conflicting data also challenged the candidate gene analysis, producing only weak transcript differences between the prickled and prickle-free phenotypes, probably due to the glandular structures observed that are similar to glandular hairs in *Rubus* and could have a similar initiation and developmental pathway.

A study utilizing prickled and a prickle-free mutant of *S. viarum* provides some information on the molecular processes behind prickle development [19]. That study concluded that the development-related TFs R2R3-MYB, MCM1-AGAMOUS-DEFICIENS-SRF box (MADS-box), Remorins (REM) and DEFORMED ROOTS AND LEAVES1 (DRL1) play a role in prickle development and provide a link between prickle development and secondary metabolism for plant adaptation. However, molecular studies in understanding prickle development in *Rubus* are lacking.

In this study, a RNA-Seq analysis was performed on prickled and prickle-free red raspberry phenotypes to understand molecular signals for prickle development and identify candidate genes controlling the prickle-free trait. This knowledge could aid in the development of prickle-free versions

of economically important cultivars utilizing genome-editing techniques. With the expanding market for fresh raspberries and blackberries, prickle-free cultivars can improve the efficiency of production practices by reducing labor costs required for plant management and harvest.

## 2. Materials and Methods

### 2.1. RNA Isolation and Quality Control

The gross morphology of prickled versus prickle-free raspberries has been previously described [18] (Figure 1). There are clear morphological differences between the two types, with prickled plants having a mixture of prickles at varying developmental stages from fresh to hard lignified prickles on their stem, petioles and leaves and prickle-free plants with smooth epidermal surfaces with only microscopic simple trichomes. Prickle development begins with the development of a cell mass structure on the epidermal surface and continues through elongation and lignification. Prickle density is characteristically higher on immature stem segments, especially basal segments, which contain all the stages of prickle development.

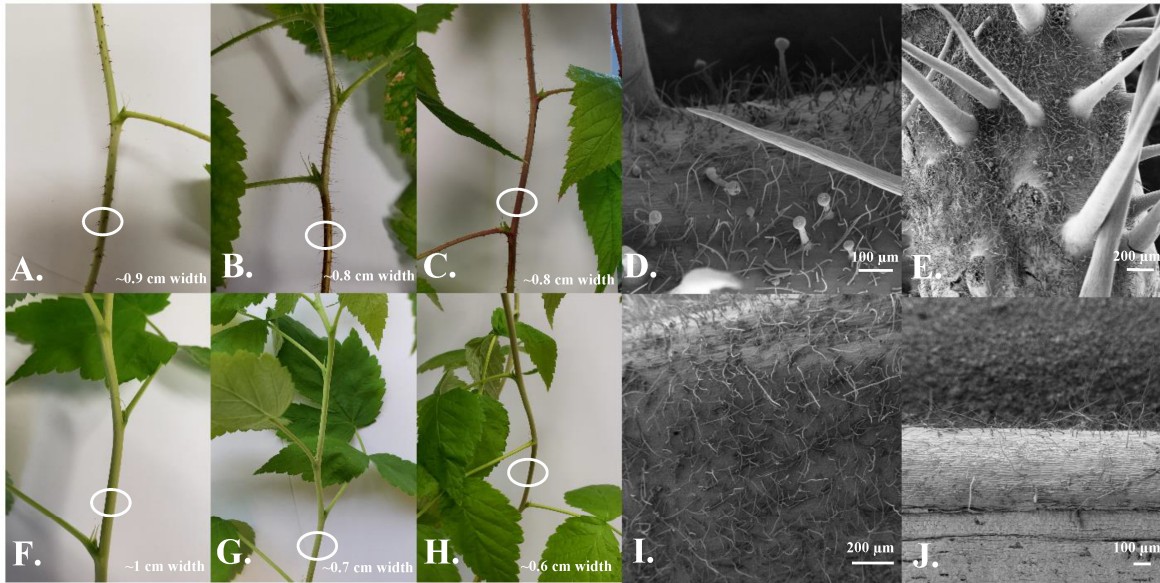

**Figure 1.** Photos showing prickles and trichomes on the surface of prickled and prickle-free *Rubus idaeus* L.: (**A–C**) prickled and (**F–H**) prickle-free. SEM micrographs showing (**D,E**) prickles, glandular trichomes and simple trichomes on prickled stems and (**I,J**) simple trichomes on prickle-free stems.

Epidermal tissue was collected from the basal segments of immature canes of three greenhouse-grown replicates of the prickled red raspberry cultivar Caroline and the prickle-free cultivar Joan J., as well as samples from 3 prickled and 3 prickle-free offspring from a previously described population between these cultivars [18]. The tissue was frozen immediately in liquid nitrogen after collection and stored at −80 °C. Total RNA was extracted using an RNeasy Plant Mini Kit (Qiagen, Valencia, CA, USA) following the manufacturer's protocols. A spectrophotometric analysis (ND1000, NanoDrop Technologies, Wilmington, DE) was used to determine the total RNA qualities and quantities.

### 2.2. RNA-Seq Assay and Illumina Sequencing

RNA-Seq libraries (3′) were prepared from ~500 ng of total RNA per sample using the Lexogen QuantSeq 3′-mRNA-Seq Library Prep Kit FWD for Illumina (https://www.lexogen.com/quantseq-3mRNAsequencing/). The libraries were quantified on a plate reader with intercalating dye and pooled

accordingly for a maximum uniformity of samples. The pool was quantified by digital PCR and sequenced on an Illumina NextSeq500 sequencer, single-end 1×86bp and demultiplexed based upon six base i7 indices using Illumina bcl2fastq2 software (version 2.17; Illumina, Inc., San Diego, CA, USA).

### 2.3. Quantitative Expression Analysis Methods

Illumina adapters were removed from the de-multiplexed fastq files using Trimmomatic version 0.36 [37]. Poly-A tails and poly-G stretches of at least 10 bases in length were then removed using the BBDuk program in the package BBMap (https://sourceforge.net/projects/bbmap/). The trimmed reads were aligned to an unpublished raspberry genome assembly (Lachesis_Heritage Version 1.1) using the STAR aligner version 2.5.3a [38]. For the following STAR indexing step for generating Sequence Alignment/Map (SAM) files, which is required for downstream analyses, the gff3 file was converted to gtf format with the gffread program from Cufflinks version 2.2.1 [39]. The output Sequence Alignment/Map (SAM) files were converted to Binary Alignment/Map (BAM) format using SAMtools version 1.6 [40]. Gene counting was then performed with HTSeq-count version 0.6.1 [41] in order to prepare the data for counting gene overlap reads in the differential expression analysis.

### 2.4. Differential Gene Expression

The R package DESeq2 version 1.20.0 [42] was used to obtain normalized counts to correct for library size and RNA composition bias and to test for differential expression between the pricked and prickle-free samples from epidermal tissue for all genes with at least 20 raw counts across all the samples. Differentially expressed genes (DEGs) were identified between samples with adjusted *p*-value ≤ 0.05 and log2-fold change >2 or <−2 as the significance threshold. Multidimensional scaling (MDS), similar to principle components analysis, was used to cluster the samples according to their overall similarity of gene expression pattern.

### 2.5. Functional Annotation Using Blast2GO/OmicsBox and GO Enrichment Analysis

Blast2GO/Omicsbox version 1.3.11 was used to associate Gene Ontology (GO) terms with individual genes from raspberries. BLASTX was performed by searching against the NCBI-nr protein database using an e-value cutoff of $10^{-3}$ and maximum number of allowed hits fixed at 20. InterProScan was run to scan sequences for matches against the InterPro protein database simultaneously using default parameters. Blast2GO function Mapping > Run GO Mapping step menu was used to perform GO mapping, followed by Blast2GO function Annotation > Run Annotation with an e-value cutoff of $10^{-6}$ was used to perform GO Annotation. Gene Ontology enrichment analysis to test for the enrichment of gene sets was carried out by comparing GO annotations from differentially expressed genes in prickle-free plants (test set) with complete GO annotations (reference set) by running Fisher's exact test using OmicsBox V1.3.11. Upregulated genes in the sample tissues were indicated by GO terms that were over or under-represented at a *p*-value of 0.05 (hypergeometric test with Benjamini and Hochberg false discovery rate correction).

### 2.6. Expression Analysis through Quantitative Reverse-Transcription PCR (qRT-PCR) Analysis

Real-time PCR was performed in 20 μL for a set of selected genes using Fast SYBR Green PCR Master Mix (Applied Biosystems Foster City, CA, USA). The list of selected genes and oligonucleotide primers (Integrated DNA Technologies Inc., Coralville, IA, USA) used for each gene is listed in Tables 1 and 2. Initially, ten genes were randomly selected in order to confirm the accuracy of the sequencing data. After confirmation, expression patterns were determined in seven highly downregulated development-related transcription factors in prickle-free plants. Expression patterns between prickled and prickle-free plants were compared. Ubiquitin was used as the internal control for normalizing the expression. The relative expression was calculated by the $2^{-\Delta\Delta Ct}$ method.

**Table 1.** List of primers used for qRT-PCR analysis for transcriptome data validation. Gene IDs are based on the annotation for an unpublished raspberry genome assembly (Lachesis_Heritage Version 1.1).

| | | Selected Primers Used for Transcriptome Data Validation | |
|---|---|---|---|
| | | **Primer Sequence 5′ to 3′** | |
| **S.N.** | **Gene IDs** | **Forward** | **Reverse** |
| 1 | 19386_g | CCCTCATAATCTCCACAGGTTT | ATTCCAGCCACTGCCATAATA |
| 2 | 3610_g | TCGTGGTGCATCAGCTTTAG | CTCCATCTTCCTGCCCATATTT |
| 3 | 4624_g | GAGGAGATTGGGATGGATGTT | CAGATGCTCCAATGCTGAAAG |
| 4 | 9394_g | CTTCTGTGATCGAATTGGGTTTG | CAGCACCACCACCTTGATAA |
| 5 | 29335_g | GCAGCTAAGGACATGGAGAAAG | GGGATATGATGATGCTGGGTTTAG |
| 6 | 21030_g | GTCAGTGACTGGTACAGGTATTT | CGATCCCTACTTTCCACCATAC |
| 7 | 18962_g | CGCATCCGGTCTTACCATTTA | TAGGCAGCATTACCGAAACTC |
| 8 | 14085_g | GCCTCTCTGTATTTCCCTATGC | GCGGAGGTTGATCGATTCTT |
| 9 | 9950_g | CTCGATACCGAACCTCCAAAG | CTCCGCAAACCCTAGCTAAA |
| 10 | 5631_g | TCATCACCGAGTCCAAACAC | GCACGGGTTTGATGAATTGG |

**Table 2.** List of primers used for expression analysis of transcription factors. Gene IDs are based on annotation for unpublished raspberry genome assembly (Lachesis_Heritage Version 1.1).

| | | | Selected Primers Used for Expression Analysis for Transcription Factors | |
|---|---|---|---|---|
| | | | **Primer Sequence 5′ to 3′** | |
| **S.N.** | **Gene IDs** | **Gene Names** | **Forward** | **Reverse** |
| 1 | 8958_g | R2R3-MYB | GCGGAGGACGGTTTGATTAG | CCACAGAAACCCTCCATGATATT |
| 2 | 3714_g | MADS-box | CAACAGCAGCAAACGAATATGA | GGTGATTGGACTCGAGGATTAC |
| 3 | 9441_g | NAC | ACGTGCTGATAACCCAGATG | CAACTCCACCAGTAGCCAAA |
| 4 | 5478_g | C2H2 | CAGTTTGCAGTGCTGTGATTAT | GCAAACTGCCCTGACAAATC |
| 5 | 13766_g | WEREWOLF | AGTTTGTGGAGCCTGATAATGA | GTGGGAAGAGTGTTAGGCTTAG |
| 6 | 19810_g | AP2/ERF | GAGGTGATAATCGGAAGCAAGA | GACCAGAAGAGCATCCCATATC |
| 7 | 8771_g | GRF5 | AGGGACGAGACGACCATATTA | GACGCCTTCTTTCTTTCTTTCTTC |

## 3. Results

### 3.1. Differentially Expressed Genes (DEGs)

A total of 3599 statistically differentially expressed genes were identified between epidermal tissues of prickle-free compared to prickled genotypes/phenotypes. Of the 3599 genes, 985 with an expression fold change <−2 and 2> were used for further functional evaluations (Figure 2A). Multidimensional scaling (MDS), similar to principle components analysis, was used to cluster the samples according to their overall similarity of gene expression pattern (Figure 2B) in order to determine if the gene expression patterns between the phenotype classes could be clearly distinguished. Samples from prickled epidermis formed one cluster and samples from prickle-free epidermis formed a second cluster.

Among the 985 genes, 233 upregulated and 752 downregulated genes were found in the epidermis of prickle-free tissue samples (Figure 2A). Several members of the R2R3-MYB family of transcriptions factors (TFs) were found to be downregulated in prickle-free epidermis. TFs MYB16-like (*Arabidopsis* MIXTA-like R2R3-MYB family member), which regulates epidermal cell outgrowth of petal conical cells and trichomes [43], was among the top downregulated genes in prickle-free epidermis. Several other TFs, including growth-regulating factors 5 and 3 (GRF5 and GRF3); TF WEREWOLF (WER); MADS-box TFs; R2R3-MYB TFs MYB8 and MYB111; NAM, ATAF1/2 and CUC2 (NAC) and APETALA2/ETHYLENE RESPONSIVE FACTOR (AP2/ERF) were also among the top downregulated genes in the prickle-free epidermis (Table 3).

**Number of up and downregulated genes**

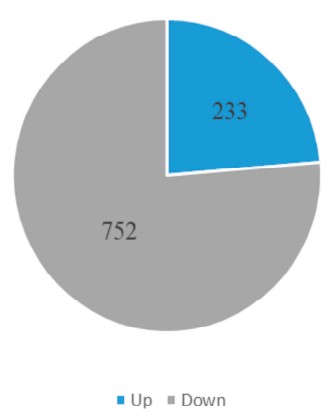

■ Up  ■ Down

**A.**

**MDS Plot**

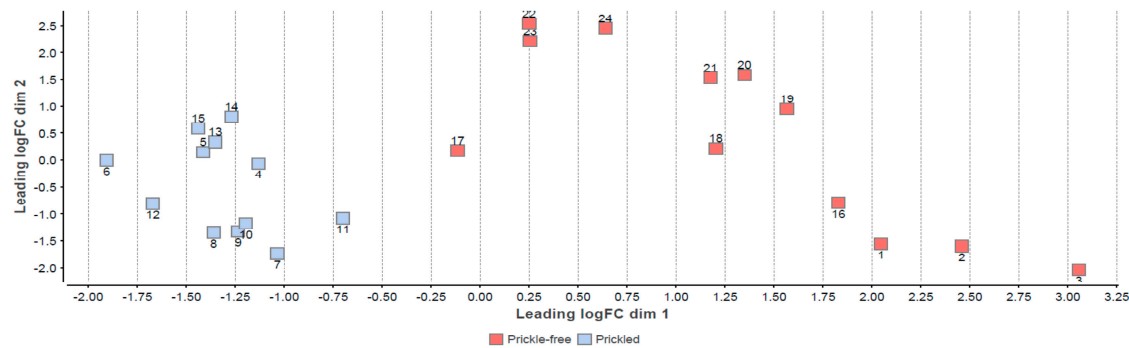

**B.**

**Figure 2.** (**A**) Number of up and downregulated genes in the epidermis of prickle-free plants. (**B**) Multidimensional scaling plot clustering samples based on their overall similarity of gene expression patterns. This plot was created using R package DESeq2 version 1.20.0. Prickled samples formed one cluster, and prickle-free sample formed another cluster, clearly distinguishing gene expression patterns between the phenotype classes. Numbers near red and blue boxes represent the sample numbers. Axes represent first and second principal components and, therefore, have no units. Samples that are near each other in the two-dimensional space are more similar with respect to gene expressions.

**Table 3.** List of the top 15 up- and 15 downregulated genes in the epidermal tissue of prickle-free versus prickled *Rubus idaeus* L. Gene IDs are based on annotation for unpublished raspberry genome assembly (Lachesis_Heritage Version 1.1) and correspond to gene descriptions in the NCBI database. NAC: NAM, ATAF1/2 and CUC2.

| Gene IDs | Log2-Fold Change | *p*-Value (Adjusted) | Description |
|---|---|---|---|
| 4820_g | −7.73 | $1.40 \times 10^{-22}$ | Myb-related protein |
| 26462_g | −7.45 | $4.64 \times 10^{-17}$ | GDSL esterase/lipase At45670 |
| 4747_g | −7.10 | $2.15 \times 10^{-18}$ | Phylloplanin-like |
| 4737_g | −6.53 | $1.83 \times 10^{-13}$ | Phylloplanin |
| 28832_g | −6.20 | $3.55 \times 10^{-17}$ | Putative proteinase inhibitor I13, potato inhibitor I |
| 8958_g | −6.03 | $5.87 \times 10^{-5}$ | Transcription Factor Myb16-Like (*Rosa chinensis*) |
| 26472_g | −5.67 | $1.23 \times 10^{-15}$ | Transcription factor MYB8-like |
| 4741_g | −5.63 | $3.02 \times 10^{-28}$ | Very-long-chain (3R)-3-hydroxyacyl-CoA dehydratase PASTICCINO 2A-like |

**Table 3.** *Cont.*

| Gene IDs | Log2-Fold Change | *p*-Value (Adjusted) | Description |
|---|---|---|---|
| 13982_g | −5.29 | $8.87 \times 10^{-16}$ | Major latex-like protein |
| 13766_g | −5.13 | $3.46 \times 10^{-22}$ | Transcription factor WER-like |
| 23514_g | −5.10 | $5.62 \times 10^{-7}$ | Ethylene-responsive transcription factor ERF109-like |
| 3903_g | −4.76 | $8.02 \times 10^{-15}$ | NAC domain-containing protein 79-like |
| 19874_g | −4.60 | $2.25 \times 10^{-11}$ | *Rosa chinensis* proline-rich 33-kDa extensin-related protein-like |
| 3948_g | −3.26 | $5.15 \times 10^{-8}$ | *Rosa chinensis* uncharacterized LOC112189886 |
| 26292_g | −3.02 | $1.38 \times 10^{-62}$ | Uncharacterized protein LOC112197621 (*Rosa chinensis*) |
| 23873_g | 21.66 | $4.26 \times 10^{-11}$ | Putative spindle and kinetochore-associated protein |
| 22675_g | 8.48 | $6.66 \times 10^{-9}$ | Transcription factor MYB36 (*Rosa chinensis*) |
| 5242_g | 7.11 | $7.16 \times 10^{-8}$ | Uncharacterized protein LOC112181570 |
| 13029_g | 6.61 | $2.31 \times 10^{-8}$ | Uncharacterized protein LOC112167160 (*Rosa chinensis*) |
| 14750_g | 6.26 | $2.31 \times 10^{-10}$ | Protein SRG1-like (*Rosa chinensis*) |
| 12450_g | 6.23 | $6.32 \times 10^{-6}$ | Putative jacalin-like lectin domain-containing protein (*Rosa chinensis*) |
| 1056_g | 6.00 | $2.47 \times 10^{-9}$ | 1-aminocyclopropane-1-carboxylate oxidase 5-like |
| 5349_g | 5.75 | $2.68 \times 10^{-6}$ | Transcription factor RAX2-like |
| 20962_g | 5.74 | $1.31 \times 10^{-7}$ | Peroxidase 27-like |
| 18908_g | 5.55 | $3.99 \times 10^{-5}$ | Probable beta-1,3-galactosyltransferase 8 isoform X2 |
| 11647_g | 5.397 | $1.11 \times 10^{-7}$ | Probable E3 ubiquitin-protein ligase ATL44 |
| 9170_g | 5.05 | $1.08 \times 10^{-11}$ | Hypothetical protein |
| 2126_g | 4.95 | $4.48 \times 10^{-5}$ | Hypothetical protein RchiOBHm_Chr1g0330981 (*Rosa chinensis*) |
| 19143_g | 4.95 | $4.24 \times 10^{-13}$ | Transcription factor bHLH94-like (*Fragaria vesca* subsp. *vesca*) |
| 23421_g | 4.93 | $4.84 \times 10^{-5}$ | Putative plant lipid transfer protein/Par allergen (*Rosa chinensis*) |

### 3.2. GO Analysis and GO Enrichment Analysis

To further analyze the role of DEGs in prickled and prickle-fee samples, a GO analysis was performed to ascertain the function of DEGs. This analysis provided GO terms for each gene based on their functions. Blast2GO/Omicsbox version 1.3.11 was used to associate Gene Ontology (GO) terms with individual genes from raspberries. Associating sequence with GO and other functional annotations is a critical step toward enabling the analysis of high-throughput gene expression. Sequences were searched against the nr protein database and the InterPro database, and the results were imported into the Blast2GO/Omicsbox program's graphical user interface.

The results of the GO analysis were divided into groupings based on functions: biological processes, cellular components and molecular functions. This provided a visual representation of the functional characteristics of the genes based on their GO aspects. Plots summarizing GO terms and the percentage of gene products assigned to each term are shown in Figure 3. Similar to *Arabidopsis* and other plants, a large number of genes were annotated with terms related to transcription factors. Approximately 14% of transcripts received the GO annotation "DNA binding". These sequences likely represented expressed transcription factors involved in the regulation of gene expression during prickle development. Similarly, results from the GO analysis indicated that the top significantly enriched GO terms in the cellular component group was an integral component of the membrane (44%). The regulation of transcription (15%), response to stress (16%) and response to chemicals (15%) were among the GO terms indicating significantly altered biological processes in the prickle-free plants.

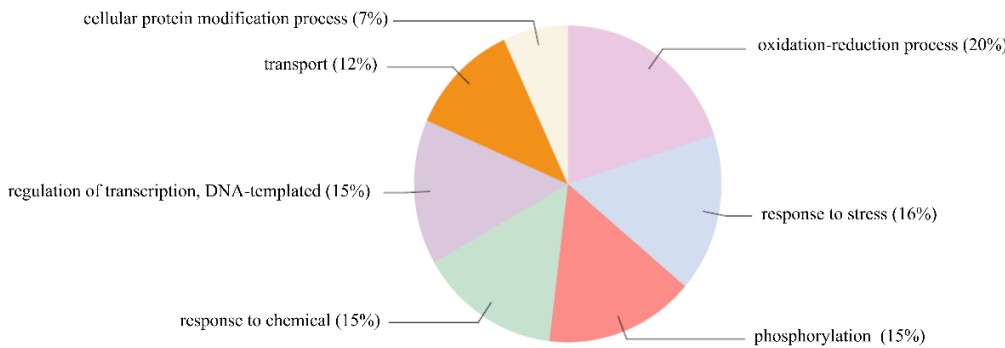

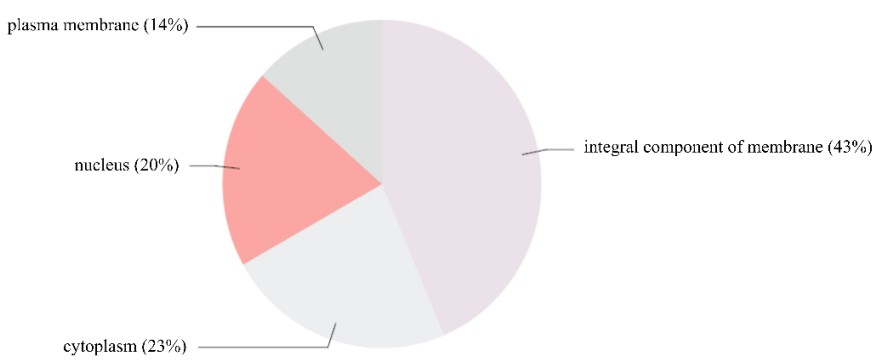

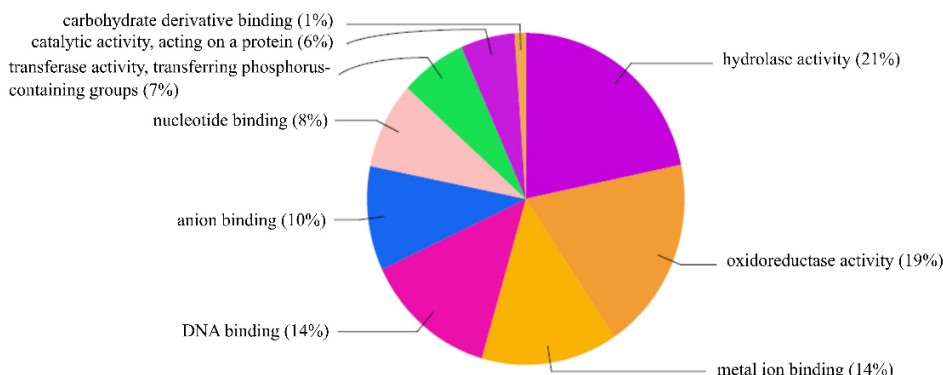

**Figure 3.** Gene Ontology annotations for the three sub-trees of Gene Ontology (GO): biological process, molecular functions and cellular components. Numbers indicate the percentage of protein-coding raspberry transcripts assigned to each category. Plots were made using Blast2GO/Omicsbox.

The frequencies of GO terms between prickle-free plants and prickled plants were then compared using GO information from all the genes in the genome. This was compared with the complete dataset using Fisher's exact test with multiple testing false discovery rate correction implemented in OmicsBox V1.3.11 [44]. The analysis showed 20 GO terms being enriched and that a significant number of DEGs were annotated with terms related to the cellular anatomical entity, catalytic activity, membrane,

intrinsic and integral component of membrane, transcription regulator activity and DNA-binding transcription factor activity (Figure 4).

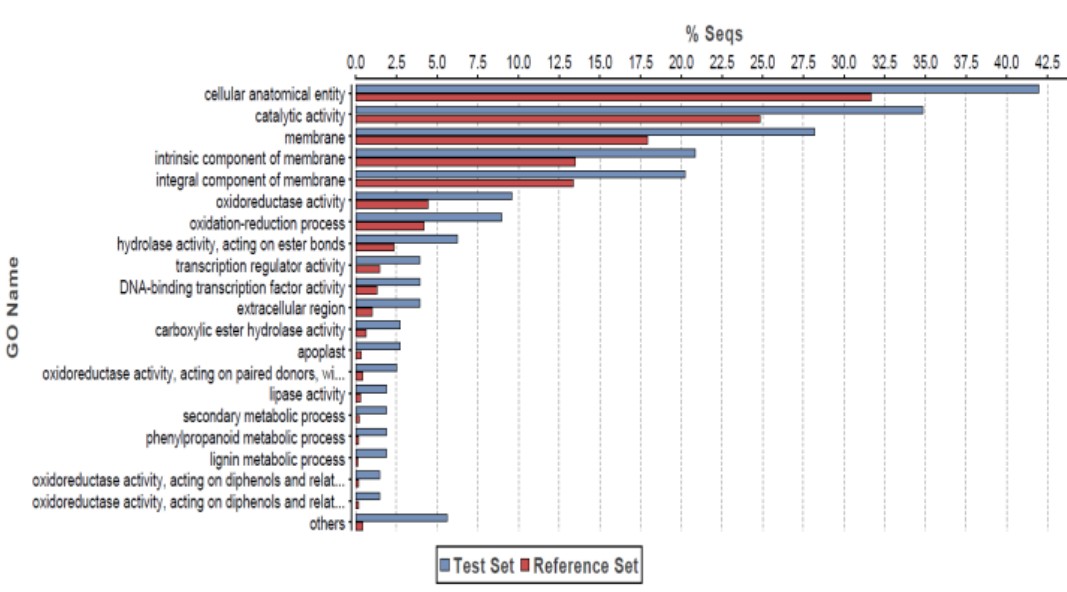

**Figure 4.** GO enrichment analysis of the differentially expressed genes using Fisher's exact test with a *p*-value cutoff applied as 0.05. The X-axis is the percentage of genes mapped by the term and represents the abundance of the GO term. The percentage for the input list is calculated by the number of genes mapped to the GO term divided by the number of all genes in the input list. The same calculation was applied to the reference list to generate its percentage. These two lists are represented using different custom colors. The Y-axis is the definitions of the GO terms.

*3.3. Putative Transcription Factors (TF) Differentially Expressed in Prickle-Free Plants*

Transcription factors (TF) play key roles in the development of plant organs [45]. In this study, 75 DEGS belonging to 15 TF families that were identified (Figure 5). Members of various TF families, such as MYB-related (6 up and 15 down), MADS-box (3 up and 2 down), homeobox (1 up and 3 down) and NAC (1 up and 8 down) were either up- or downregulated. Cysteine-2/Histidine-2 (C2H2) (1) and HGMI-Y (1) were upregulated, while AP2/ERF (8), bHLH (3), DNA-BINDING WITH ONE FINGER (DOF) (2), GATA (1), HEAT STRESS TRANSCRIPTION FACTOR (HSF) (1), RELATED TO AB13 AND VP1 (RAV) (2), TEOSINTE BRANCHED1/CYCLOIDEA/PROLIFERATING CELL FACTOR (TCP) (3), WRKY (13) and YABBY (1) were downregulated in prickle-free epidermis.

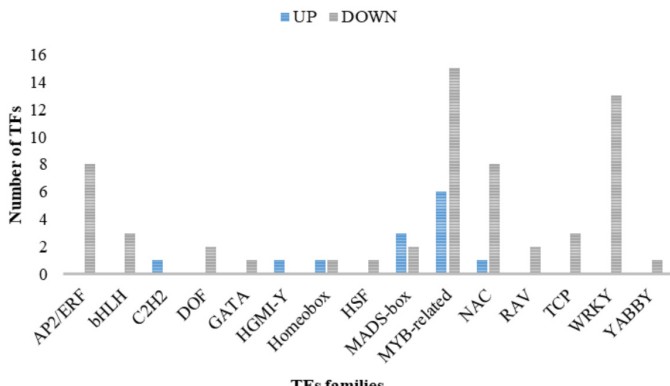

**Figure 5.** Analysis of differentially expressed transcription factors (TFs) in the epidermis of prickle-free plants in comparison to prickled plants. A total of 75 differentially expressed genes (DEGs) belonging to 15 TF families were identified.

*3.4. Validation of Transcriptome Data by qRT-PCR Analysis*

The expression patterns of 10 randomly selected genes were determined using qRT-PCR analysis to confirm the accuracy of the sequencing data. Expression patterns obtained from qRT-PCR and RNA sequencing data analysis showed a similar pattern of gene expression supporting the reliability of the data from RNA sequencing. Expression patterns of seven development-related TFs that were highly downregulated in the prickle-free epidermis were also compared in prickled and prickle-free plants. qRT-PCR revealed the downregulation of the development-related TFs in prickle-free plants similar to the results of DEG analysis (Figure 6).

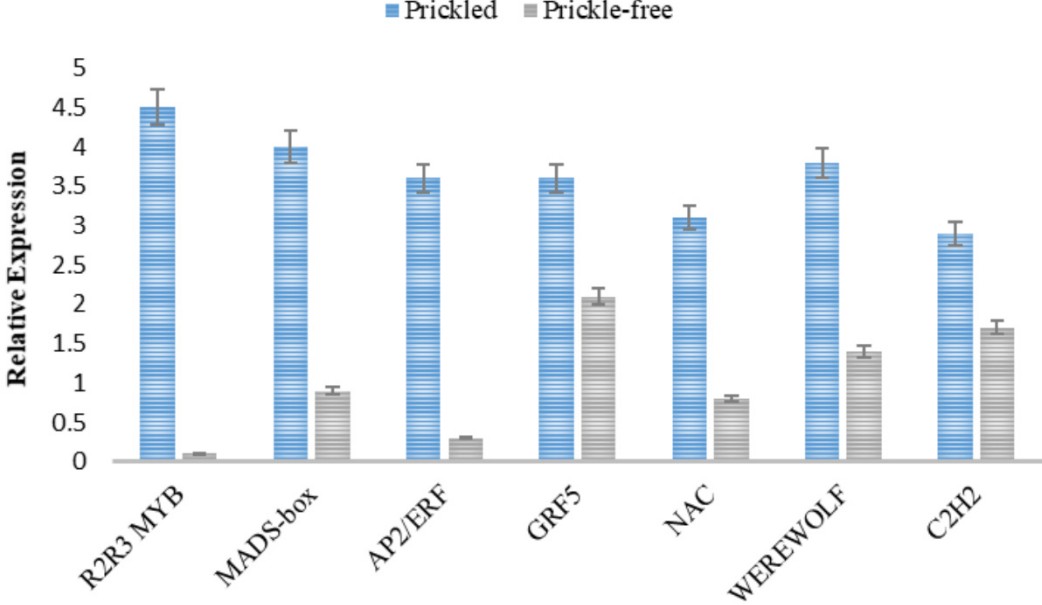

**Figure 6.** Relative expression patterns of plant development-related transcription factors (TFs) in prickled and prickle-free epidermis. Error bars represent SE of means of triplicates.

## 4. Discussion

Previous studies have shown that prickle development in *Rubus* species is associated with presence or absence of glandular structures resembling trichomes early in development [16,18]. Glandular or secreting trichomes are present in many vascular plants and are multicellular structures with secreting glands at the tip of the stalk which often produce and store terpenoids, phenylpropanoid oils and other secondary metabolites [46,47]. The transcriptional network regulating the development and patterning of unicellular trichomes is well characterized in *Arabidopsis*, however, understanding the development of glandular trichomes is cursory at this stage [20–22]. Simple trichomes are present from the very beginning of seed germination and can be observed at the surface of the first cotyledon. In red raspberry, proto-prickles can be observed on the fringe of cotyledons and are diagnostic for the presense of mature prickles later in development. Mature prickle development begins 10 to 15 days after seed germination. Moreover, prickle development varies through the life stages of the plant with basal stem tissue developing higher density compared to later stages in development.

Glandular trichomes also produce and store a large number of secondary metabolites [46,48] including flavonoids, alkaloids and terpenes, to name a few. Secondary metabolites are known to defend plants against both biotic (herbivores and pathogenic microorganisms) and abiotic stresses [49]. In a recent study, several genes associated with secondary metabolism were found to be significantly up-regulated in a prickle-free mutant of *S. viarum* compared to the prickled counterpart. Functional enrichment analysis showed that siginificantly altered biological processes in the prickle-free genotype included the GO terms response to biotic and abiotic stimulus (GO:0009628 and GO:0009607), response to defense (GO:0006952) and response to stress (GO:0006950). Significantly altered molecular function

included GO terms related to catalytic activity (GO:0003824), binding (GO:0005488) and transporter activity (GO:0005215), while top GO terms related to cellular component inlcuded cell and cell parts (GO:0005575). Since siginifically altered biological processes consisted most of the plant defense and stress related processes, it was hypothesized that, under the absence of one defense mechanism (prickles), the plants could potentially use secondary metabolites as defense mechanism against abiotic stresses [19]. In our study, results from GO analysis identified regulation of transcription (GO:0006355), response to stress (GO:0006950) and response to chemical (GO:0042221) among the top significantly altered biological processes in the prickle-free plants. However, no increased expression of genes involved in secondary metabolism in prickle-free plants was observed. This may be due to prickle-free *Rubus* complelety lacking glandular trichomes or any cell mass structure where such metabolites would be produced as opposed to prickle-free mutant of *S. viarum* where glandular trichomes were still present in the absence of prickles. Top GO annotation related to cellular component inlcuded integral component of cell membrane (GO:0005887) and transcripts with GO annotation "DNA binding" (GO:0003677), related to transcription factors and transferase activity were among top significantly altered molecular function.

Gene Ontology enrichment analysis of genes differentially expressed between prickled and prickle-free epidermis found that a significant number of DEGs were annotated with terms related to cellular anatomical entity, transcription regulator activity, DNA-binding transcription factor activity, and catalytic activity. GO for cellular anatomical entity included trichome apex, trichome branch and trichome tip under its GO tree among several other cellular components, thus significantly altered DEGs. The next two significant GO annotations were "transcription regulator activity" and "DNA-binding transcription factor activity". These terms suggest the involvement of transcription factors in prickle development. In addition, another significantly altered GO term between prickled and prickle-free epidermis was catalytic activity suggesting the prickle development process might also be rich in biosynthetic processes which is lacking in the prickle-free genotype.

In *Arabidopsis*, the transcriptional network regulating the development and patterning of unicellular trichomes is well characterized. More than 30 genes have been identified that contribute to different aspects of trichome development in *Arabidopsis* [23,27]. Unicellular trichome cell fate is intiated by a complex of members of three TF gene families: MYB, bHLH, and WD40 [50]. In this case, the MYB family protein is GLABRA1 (GL1), the bHLH family protein is GLABRA3 (GL3) or its redundant partner ENHANCER OF GL3 (EGL3), and the WD40 family protein is TTG1 [24,50–53]. The complex of these three act in concert to activate trichome initiation and patterning. TTG1, GL1, and GL3 or EGL3 help in inducing the expression of *GLABRA 2* (*GL2)*, a positive regulatory gene of trichome formation, by forming an activator complex TTG1–GL3/EGL3–GL1 [54]. In contrast, R3 MYBs including TRYPTICHON (TRY) [55], CAPRICE (CPC) [56], TRICHOMELESS1 (TCL1) [57], TCL2 [58,59], ENHANCER OF TRY AND CPC1 (ETC1) [60,61], ETC2 [62] and ETC3 [59,63] negatively regulate trichome formation by binding with GL3 or EGL3 competing against GL1 and thus blocking the activator complex necessary for trichome initiation [64–69]. Similarly, functional characterization of a WD-repeat homolog of CsTTG1 in cucumber, which plays an important role in the formation of cucumber fruit bloom trichome and warts, revealed that it is mainly expressed in the epidermis of the ovary and that its overexpression in cucumber alters the density of fruit bloom trichomes and spines [28]. Morerver, silencing its expression leads to inhibition of the initiation of fruit spines.

Our RNA sequencing results revealed down-regulation of several transcription factors belonging to different families: R2R3-MYB, homeobox, C2H2, AP2/ERF, MADS-box, and NAC in the epidermis of prickle-free plants. The expression pattern obtained through qRT-PCR analysis showed significant down-regulation of R2R3-MYB, AP2/ERF, MADS-box, and NAC in prickle-free epidermis in comparison to prickled epidermis confirming our results from RNA-Seq analysis. A transcription factor, GL1 belonging to R2R3-MYB family has been determined to function in trichome development via incorporation into the MYB-bHLH-WD40 regulatory complexes as discussed earlier [53].

The R2R3-MYB subfamily is the largest in plants, and it is characterized by two types of R domain in the N-terminal end and usually by a transcription activator or repressor in the C-terminal end [70]. Most R2R3-MYB proteins have been found to be involved in the control of developmental processes, primary and secondary metabolism, cell fate and identity, and responses to biotic and abiotic stresses [70]. The WD40–bHLH–MYB transcription complex provides an interesting example, where different MYB proteins determine the specific function. For instance, in vegetative tissues, the complex regulates anthocyanin biosynthesis when MYB75/90/113/114 is incorporated [71]. However, the complex regulates trichome initiation and branching when GL1/MYB23 is recruited [50,72], and trichome development when MYB82 is recruited [73]. Similarly, in roots, the complex regulates root hair patterning when WER is incorporated [74].

In present study, one of the top down-regulated genes in prickle-free epidermis is MIXTA-like R2R3-MYB family member. Members of the MIXTA-like family are known to promote conical cell outgrowth and trichome initiation in diverse plant species [43]. Contribution of MIXTA-like R2R3-MYB family members in trichome development has been identified in other plants like snapdragon (*Antirrhinum majus* L.) and its homologs in *Arabidopsis*, cucumber, cotton (*Gossypium hirsutum* L.) and many other species [70,75–84]. In cucumber, a MIXTA-like homologs *CsMYB6* has been determined to regulate epidermal cell differentiation, cuticle wax biosynthesis, and trichome morphogenesis [85–88]. A study conducted to identify and characterize the genome-wide R2R3-MYB family in three species in the Rosaceae family: *Malus domestica* Borkh. *Prunus persica* (L.) Batsch and *Fragaria vesca* L., identified 44 functional subgroups with seven unique to Rosaceae family [89]. Functional analysis of the TFs were performed based on the clustering of R2R3-MYB genes of *Arabidopsis*. The study identified two R2R3-MYB, *Arabidopsis* transcription factor *AtMYB5* in subgroup 16 and *AtMYB106/NOK* and *AtMYB16*/MIXTA in subgroup 5. *AtMYB5* is known to regulate trichome morphogenesis and mucilage synthesis [90]. *AtMYB106/NOK* and *AtMYB16*/MIXTA are known to participate in trichome development [76]. Therefore, MIXTA-like R2R3-MYB family member could be one of the key regulators of prickle development.

Very few studies have been done to understand molecular aspects of prickle development in any species so far, the closest one being prickle development in *S. viarum* [19]. This study concluded that the development related transcription factors R2R3-MYB, MADS-box, REM and DRL1 play a role in prickle development. The results in the present study are comparable to that on *S. viarum* in that the down regulation of development related TFs like MIXTA-like R2R3- MYB and MADS-box was observed in the prickle-free epidermis of *Rubus*. Moreover, a direct link between MIXTA-like R2R3-MYB family members to trichome development has been previously established, making this TF member a potential candidate gene for prickle formation in *Rubus*. Similarly, other essential development related TFs identified in this study include AP2/ERF, MADS-box, and NAC. The AP2/ERF transcription factors consist of a large gene family coding proteins that are characterized by the presence of an AP2 domain, which directly binds to the GCC box or DREB/C-repeat elements at the promoter of downstream target genes [91]. AP2/ERF proteins are classified into four main subfamilies: AP2, RAV (related to ABI3/VP1), DREB (dehydration-responsive element binding protein), and ERF, according to the number and similarity of the AP2 domain. The AP2/ERF transcription factors are known to regulate diverse processes of plant development such as vegetative and reproductive development and cell proliferation, as well as abiotic and biotic stress responses, and plant hormone responses [92–94]. In *Arabidopsis* cotyledons, the embryonic identity was disturbed leading to ectopic trichome formation when *EMBRYOMAKER* (*EMK*), which belongs to the AP2 subfamily, was overexpressed [95]. Similarly, in maize (*Zea mays* L.), the *Glossy15* gene encoding a protein containing two AP2 domains, is required for the expression of juvenile epidermal traits, including leaf trichomes [96]. Moreover, another AP2/ERF TF, Hairy Leaf 6 (HL6), interacts with a homeodomain-containing the protein OsWOX3B, and regulates trichome formation in rice [97]. However, the specific roles that AP2/ERF transcription factors play in trichome development are still unclear.

MADS-box genes are the key members of regulatory networks behind multiple developmental pathways. These genes regulate the networks involved in plant responses to stress as well as the plant developmental plasticity response to seasonal fluctuations [98]. In plants these genes play central roles in flower and fruit development [99,100]. Other MADS-box genes are expressed in vegetative tissues, ovules and embryos, suggesting that this family of genes plays diverse roles throughout plant development [101–104]. In terms of trichomes, *AGL16*, a MADS-Box gene, is expressed in mature guard cells and trichomes found in both the abaxial and adaxial epidermis suggesting their evolution beyond flowers [105]. NAC transcription factors have several biological roles in regulation of plant growth and development including the formation of shoot apical meristems, reproduction, floral organ development, lateral shoot development, and defense response against biotic and abiotic stresses [106]. Recently it was discovered that the plant-specific NAC family of TFs could play an important role in regulating fruit spine initiation and development in cucumber [107]. The study performed a comprehensive analysis and identified a set of NAC genes that represents targets for future studies of cucumber fruit spine development. Out of 91 NAC genes divided into 6 subfamilies, 6 genes were identified to play a positive regulatory roles in fruit spine development.

## 5. Conclusions

RNA-Seq is a powerful tool for elucidating the genetic control of complex traits at a specific time point during development. This study examines the differences between the genes expressed in epidermal tissues of prickled and prickle-free red raspberry genotypes utilizing RNA-Seq analysis and qRT-PCR validation. The differential expression analysis revealed the significant downregulation of some vital development-related transcription factors (TFs), including a MIXTA-like R2R3-MYB family member, MADS-box, AP2/ERF, NAC in prickle-free epidermis tissue, which were confirmed by the qRT-PCR analysis. In conclusion, this study provides an insight on the potential genes responsible for prickle production in *Rubus* and is a step forward in the understanding of prickle formation processes and molecular genetic studies in *Rubus*. To follow up, a GBS analysis to locate the prickle-free locus will be utilized with the RNA-Seq data to further examine the genomic location potential candidate genes in conjunction with phenotypic data from a segregating population. A clustered, regularly interspaced short palindromic repeats/CRISPR-associated protein 9 (CRISPR/Cas9) knockout approach can then be used to confirm the gene responsible for the prickle-free phenotype.

**Author Contributions:** A.K. conducted the research in this manuscript as partial requirement for her Ph.D. degree program at Cornell University under the advisement of C.A.W. who contributed to the project design, data analysis and interpretation, and editorial guidance. All authors have read and agreed to the published version of the manuscript.

**Funding:** This research received no external funding.

**Conflicts of Interest:** The authors declare no conflict of interest. The funders had no role in the design of the study; in the collection, analyses, or interpretation of data; in writing of the manuscript, or the decision to publish the results.

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
