# Peer review of "RNA-Seq Analysis of Prickled and Prickle-Free Epidermis Provides Insight into the Genetics of Prickle Development in Red Raspberry (Rubus ideaus L.)"

_agronomy, doi:10.3390/agronomy10121904_

Round 1

Reviewer 1 Report

The manuscript under review is devoted to the problem of prickle development in an important agricultural crop, red raspberry, where prickles/spines can impact the commercial value of the fruits and prickle-free mutants are available. The authors used a transcriptomic approach, RNAseq plus qRT-PCR, to identify putative candidate genes responsible for prickle development which eventually can be switched-off or on in order to speed breeding practices to obtain prickle-free varieties. The authors used two cultivars with opposite phenotype, prickled and prickle-free plants, the proper number of biological replicates and a well-planned experimental design. What is missing in the methodological approach is a clear statment about the developmental stage of plants and/or epidermal samples. Sampling is crucial in transcriptomic studies. The authors well described the developmental process of prickles in Discussion session, showing how prickle development varies through the life stage of the plant and at the onset of maturity after seed germination. To provide reliable expression data, samples from different plants have to be collected in a consistent way and in a comparable stage. The authors have to fix this issue in 'Materials and methods' section, and modify 'Results' and 'Discussion' accordingly. The paper presents families of genes which result to be up- or down-regulated in prickle-free samples. The authors picked up few of them to confirm their differential expression by qRT-PCR. The results are interesting and give insights at molecular level, without any doubt. Nevertheless, they didn't identify any specific gene responsible for the trait, but candidate TFs and especially families of genes. They have to adequate the title of the paper to that, and discuss more the findings of their candidate TFs and/or gene families, instead of supporting the findings explaining what other authors found in related species. To that reason, in the presented form the article can be recommended for publication in Agronomy journal, after proper changes to its title and an adequate revision of the discussion contents, together with the fixation of the sampling issue. The reviewer recommends also to correct some spelling or typo errors (see: p. 3 § 2.2; p. 6 caption of Figure 2.A; p. 7 § 3.2; p. 11 § 4; ...) and to set legends of Figure 2.A (add the meaning of numbers close to blue/red squares) and Figure 4 (Input list of caption has to be cosistent to Test set in the legend, change one or the other).

Author Response

Regarding comments from Reviewer #1: A more in depth description of the plant material was included in the materials and methods describing the age and location of tissue collected (page 3, lines 105-108). This had already been taken into account in the results and discussion so no tissue age/location related comments were made as prickles in red raspberry appear on all plant organs as stated and described previously.

Typo/spelling errors were not clear to the authors but minor adjustments were made in the manuscript. The title has been changed to more accurately reflect the information in the manuscript. Additional information was included in the discussion section to clarify the transcription factor families and their relationship to various developmental processes in different plant species and their potential role in red raspberry. Page 12, lines 407-415, 419-424, 439-462.

The figure legend for Figures 2 and 4 have been modified to more clearly describe the figures.

Reviewer 2 Report

The authors identified genes involved in prickle development in the red raspberry cultivars with or without prickle by RNA-seq analysis. As a result, several TFs including MIXTA-like R2R3-MYB family member, MADS-box, AP2/ERF and NAC domain protein were down-regulated in prickle-free epidermis tissue.

This manuscript is well-studied concerning TFs related to pickle development and the results exhibited here are reliable. However, unfortunately unless the authors performed GO analysis of DEG as compared both phenotypes, the description is only about TFs in text.

As the authors referred in Discussion part, Pandey et al. (2018) reported transcriptional analysis of prickle development in Solanum viarum Dunal. They exhibited that prickleless mutant were down-regulated the gene expression of the same kind TFs and transcript levels of genes related to hormone metabolism, secondary metabolism, biotic & abiotic stress-related were significantly changed. Therefore, the author should compare the GO analysis of this study with the previous report and discuss whether same or not.

Another minor revised point is that scales are needed in Fig.1A-C, F-H and in Fig.1 DE,IJ the scales are distinct. Change font size and color.

Author Response

Regarding comments from Reviewer #2: Additional discussion regarding the GO analysis was added to clarify these results. Page 11, lines 350-380. This more accurately described the classification of the various genes identified and the relationship to developmental processes that they describe. Figure 1 has been modified with larger font and white lettering as well as with relative scaling for the raspberry cane diameters in the gross morphology photos.